# Lin28 Regulates Cancer Cell Stemness for Tumour Progression

**DOI:** 10.3390/cancers14194640

**Published:** 2022-09-24

**Authors:** Zhuohui Lin, Mariia Radaeva, Artem Cherkasov, Xuesen Dong

**Affiliations:** 1Department of Microbiology and Immunology, University of British Columbia, Vancouver, BC V6T 1Z4, Canada; 2Faculty of Food and Land Systems, University of British Columbia, Vancouver, BC V6T 1Z4, Canada; 3The Vancouver Prostate Centre, Department of Urologic Sciences, University of British Columbia, Vancouver, BC V6H 3Z6, Canada

**Keywords:** Lin28, let-7, metastasis, metabolism, epithelial-to-mesenchymal transition, prostate cancer, cancer stem cell, Lin28 inhibitors

## Abstract

**Simple Summary:**

Cancer stem cells (CSCs) are a small population of tumour cells bearing stemness characteristics. They have been deemed as a root of cancer development and a promising drug target for anticancer therapy. Herein, we discuss the roles of an RNA-binding protein, Lin28, in regulating cancer cell stemness to drive tumour progression. Lin28 acts on various types of RNAs in cancer cells and regulates these RNA functions to control the expression of oncogenes. In these ways, Lin28 promotes cancer cell survival, growth, and invasion. We also discuss recent efforts in developing Lin28 inhibitors targeting CSCs in tumours.

**Abstract:**

Tumours develop therapy resistance through complex mechanisms, one of which is that cancer stem cell (CSC) populations within the tumours present self-renewable capability and phenotypical plasticity to endure therapy-induced stress conditions and allow tumour progression to the therapy-resistant state. Developing therapeutic strategies to cope with CSCs requires a thorough understanding of the critical drivers and molecular mechanisms underlying the aforementioned processes. One such hub regulator of stemness is Lin28, an RNA-binding protein. Lin28 blocks the synthesis of let-7, a tumour-suppressor microRNA, and acts as a global regulator of cell differentiation and proliferation. Lin28also targets messenger RNAs and regulates protein translation. In this review, we explain the role of the Lin28/let-7 axis in establishing stemness, epithelial-to-mesenchymal transition, and glucose metabolism reprogramming. We also highlight the role of Lin28 in therapy-resistant prostate cancer progression and discuss the emergence of Lin28-targeted therapeutics and screening methods.

## 1. Introduction

Cancers such as prostate cancer (PC) contain heterogeneous cell populations and, therefore, respond to anticancer treatments differently [1]. This heterogeneity is reflected not only in genomics and transcriptomics but also RNA processing re-programming that can give rise to tumour cells with various co-existing phenotypes and histological presentations [2,3,4]. When a targeted therapy is applied to a tumour, it will be effective in suppressing some populations of tumour cells and, at the same time, may provide an opportunity to allow other cell populations to thrive and eventually develop into therapy-resistant tumours. One good example is how androgen receptor (AR)-targeted therapy in metastatic PC can result in short-term tumour suppression effects but also induces castrate-resistant PC (CRPC) in the long term that presents an AR-indifferent phenotype containing weakly expressed AR and AR-regulated PSA, cancer stem cell (CSC) markers (e.g., CD44, CD133, BMI1, EZH2), and even AR negative neuroendocrine PC (NEPC) markers (e.g., CHGA, SYP) [5,6,7]. Several studies using patient tumour samples, patient-derived xenograft (PDX), and even cell models have demonstrated that there exist landscape switches in phenotypes between primary and therapy-resistant tumour cells, indicating that therapy-resistant tumours are under the control of different oncogenic signal networks than primary tumours [8,9,10]. These findings underscore that tumour heterogeneity is an inherent challenge for targeted therapies, and the identification of the driver genes of therapy-resistant tumours could inform the design of more effective treatments for cancer patients.

Cancer stem cells (CSCs) are an important component of tumour heterogeneity and have been proposed to relate to therapy-resistant tumour progression through two possible mechanisms. One is that a low number of CSCs pre-exist in primary tumours [11]. They are surrounded by a large number of relatively fast-growing non-CSC cells and remain in a quiescence/dormancy state. While therapy-induced stress is efficient to abolish non-CSCs, it can activate CSCs to give rise to tumour cells that are resistant to therapies. Single-cell sequencing of patient PC samples has shown that CRPC cells with characteristic CSC and basal cell phenotypes pre-exist in primary PC [12]. These cells express high levels of EZH2 and SYP and low levels of AR and AR-regulated PCA3 and have Rb1 loss, all of which are consistent with the molecular profile of classic late-stage NEPC [12]. Our lab also reported that the NEPC driver gene, SRRM4, is expressed in 16% of hormone-naïve primary tumours, and SRRM4-positive tumour cells increased to ~30% of tumours under AR target therapy, supporting the idea that CRPC tumour cells are pre-composite in untreated PC and become prevalent upon anticancer treatment [13]. The other mechanism is that PC cells with specific genomic features can acquire CSC phenotypes upon exposure to therapy-induced stress [14,15]. Androgen receptor (AR)-mediated signalling is essential to maintain the luminal epithelial phenotype of prostate adenocarcinoma. However, it has been reported in transgenic mouse models that, when AR inhibitors were used to treat PC cells with TP53 and RB1 gene loss, these cells gained CSC, epithelial-to-mesenchymal, basal, and neuroendocrine phenotypes through the action of transcriptional factors, such as SOX2, and, later, developed into NEPC [15]. In vitro LNCaP PC cells were reported to undergo transdifferentiation to gain CSC phenotypes when treated with AR inhibitors [14]. Gain of function of SRRM4 in DU145 cells that had TP53 and RB1 gene disruptions, but not LNCaP cells, which were TP53- and RB1-intact, induced a classic CSC gene signature, primarily driven by the Lin28–SOX2 axis [16]. These findings indicate that prostate adenocarcinoma cells are plastic and can alter their phenotypes to cope with therapy-induced stress, during which process the CSC gene network plays a key role.

Lin28A and Lin28B (from here referred to as Lin28) are RNA-binding proteins and the key drivers of CSC phenotypes [17]. Lin28, Oct4, Sox2, and Nanog are the four key reprogramming factors that induce pluripotency, with Lin28 being the only RNA-binding protein [18,19]. Lin28 is weakly expressed in normal differentiated cells but dysregulated in cancer cells, resulting in the initiation of a cascade of reactions leading to cancer cells gaining stemness, mobility, and enhanced therapy resistance [20]. Importantly, overexpression of Lin28 downregulates levels of let-7 microRNAs (miRNAs), a class of tumour-suppressor miRNA that regulates a wide range of cancer-related processes, including cell growth and proliferation, epithelial-to-mesenchymal transition (EMT), stemness, and glucose metabolism reprogramming [20]. Herein, we review the roles of Lin28 in different types of cancer and discuss the roles of Lin28 in CSC and EMT phenotype development, as well as glucose reprogramming. We also provide an overview of the impact of Lin28 on PC, since it has emerged as a new diagnostic and therapeutic target of PC [21]. Finally, we summarize advances in Lin28 inhibition with small molecules aimed at combating cancer stemness and metastasis. The hub nature of Lin28 protein makes it an attractive and promising target for chemotherapy.

## 2. Lin28/let-7 axis

### 2.1. Lin28 Structure and Function

There are two paralogs of Lin28 protein in vertebrates, Lin28A and Lin28B [22]. Their structures share high homology in protein sequences, except that Lin28B has a longer C-terminus containing the nuclear localization signal (NLS) and nucleolar localization signal (NoLS) [22,23]. Indeed, various studies have confirmed that the cellular localization of the two isoforms is different. Lin28B can be localized throughout the cells, even inside the nucleolus, while Lin28A is predominantly localized at the cytosol [22,23]. Both paralogs contain highly conserved RNA binding regions: the N terminal cold-shock domain (CSD) and a C terminal cysteine–cysteine–histidine–cystine (CCHC)-conserved zinc knuckle domain (ZKD) [22]. Lin28A and B bind the immature forms of let-7 microRNA family members and prevent them from being synthesized. Lin28A and Lin28B are mutually exclusive in expression in human cancer cell lines [24]. For instance, Piskounova et al. [24] confirmed that only Lin28A was detected in MES and IGROVE1, while only Lin28B was detected in HEK293, H1299, HepG2, and K562 cell lines. However, a minority of cancers, including ovarian cancer, germ cell tumours, and teratomas, express both paralogs of Lin28 [25,26,27].

### 2.2. Let-7 Biogenesis

Let-7 miRNAs are key regulators of embryonic development and cancer progression. They are synthesized through several steps (Figure 1). First, the primary miRNA transcripts of let-7s (pri-let-7s) are transcribed from the MIRLET7 genes by RNA polymerase II [28,29]. These transcripts form structures of hairpin stem loops consisting of a stem region, and they have a characteristic pre-element (preE) bulge and a preE loop [29,30]. Second, pri-let-7 is cleaved at the stem-loop structure of the preE by Drosha (a nuclear-localized RNase III) and Pasha (double-stranded RNA binding protein) into a shorter (60–80 nts) pre-let-7 hairpin structure [29]. The pre-miRNA is then exported out of the nucleus by exportin-5 [31]. Finally, the preE element of pre-let-7 is cleaved by Dicer to produce a mature let-7 miRNA duplex with a 3’ overhang in the cytoplasm [30]. Some members of the let-7 family (pre-let-7a-2, -7c, and -7e) contain a bulged adenosine/uridine residue at the 3’ end that impairs them from being recognized by cytoplasmic Dicer complex [32,33]. In these cases, a terminal uridylyl transferase (TUT2, TUT4, or TUT7) will mono-uridylate the 3’ overhang of pre-let-7 to allow Dicer cleavage [32,33,34]. At the cytoplasm, HIV-1 TAR RNA-binding protein (TRBP) interacts with Dicer and recruits argonaute protein (i.e., AGO2) to form RNA-induced silencing complex (RISC) [35]. One strand of the mature let-7 duplex is preferentially incorporated into the active parts of the RISC [23,36]. Besides the conventional described maturation process, Drosha/Dicer-independent biogenesis of let-7 microRNA also exists [37,38].

### 2.3. Lin28 Regulates let-7 Maturation

As shown in Figure 1, two paralogs of Lin28 bind to let-7 to prevent it from being further processed [39,40]. The two conserved domains, CSD and ZKD, have a high affinity to let-7 precursors at the terminal loop region of preE [30]. Specifically, ZKD binds to the 3’ end of the preE bulge containing the GGAG motif, while CSD preferentially binds to the preE loop with the NGNGAYNNN motif [30]. Mayr et al. [41] showed that CSD binds to microRNA first and then induces a conformational change in the preE, making it available for ZKD binding. Although CSD has a higher affinity to miRNA, ZKD binding to the sequence is required and sufficient to recruit TUTase enzymes and induce oligouridylation [42]. In the cytosol, the Lin28B CSD recognizes the preE loop of pre-let-7 while allowing ZKD to bind with the conserved motif preE bulge within let-7 (GGAG) [22,32]. ZKD binding to miRNA blocks the Dicer cleavage site, which is proximal to the GGAG-conserved preE bulge, and prevents pre-let-7 maturation [22]. It is worth mentioning that some members of the let-7 family can escape Lin28-mediated inhibition [43]. For instance, Triboulet et al. [43] confirmed that human let-7a-3 and its murine ortholog, mouse let-7c-2, bypass Lin28A-mediated repression. However, these pathways must be less common because most of the studies confirm that let-7 levels are tightly regulated by Lin28A and Lin28B. 

Lin28A and Lin28B inhibit let-7 maturation in distinct cellular compartments [40]. Lin28A is mostly found in the cytosol while Lin28B is mostly found in the nucleus, as Lin28B has an NLS [24]. In the nucleus, RNA-binding protein Musashi (MSI1) induces Lin28B nuclear localization and aids ZKD recognition of the terminal loop of pri-let-7, inhibiting Drosha cleavage [40,44]. Despite the different localization of Lin28A and B in cells, some other studies have proposed that the localization of the Lin28A and B paralogs might be cell cycle-dependent [45]. Furthermore, Piskounova et al. [24] found that, while Lin28A requires TUTase to inhibit let-7 maturation, Lin28B acts in a TUTase-independent manner, recruiting DGCR8 microprocessor instead. 

## 3. The Role of Lin28 in Cancers

The roles of Lin28A and Lin28B in cancer development and progression have been elucidated over the last decade. Numerous studies have found that overexpression of Lin28A/B leads to reduced levels of the let-7 members that stimulate tumour cells to gain stemness and invasiveness. Not surprisingly, clinical studies have confirmed that Lin28 overexpression correlates with higher chances of metastasis and overall poor survival rates, and we have summarized the main findings on Lin28 in various cancer types in Table 1. The broad range of cancer types and cancer cell lines that rely on Lin28/let-7 regulation indicates a central role for the pathway in cancer biology. Indeed, a comprehensive analysis of Lin28A and Lin28B expression over a panel of 527 human cancer cell lines revealed that the paralogs are overexpressed in roughly 15% of cancer types [25]. The plethora of Lin28A/B functions is explained by the abundance of downstream genes, including direct targets of let-7, Sox2, HMGA2, and c-Myc [17]. Furthermore, various studies have shown that therapeutic inhibition, gene knockout or RNA silencing of Lin28 reverse the stemness and invasive phenotypes (Table 1), highlighting that aberrant Lin28 expression is functional in the control of tumour cell aggressiveness. Additionally, overexpression of Lin28B in transgenic murine models was sufficient to induce hepatoblastoma and hepatocellular carcinoma, while deletion or silencing of Lin28B gene prolonged survival [46]. These studies collectively demonstrate the key importance of Lin28A/B in the development and progression of cancers.

## 4. The Role of Lin28 in Regulating the CSC Phenotype

Early evidence of the key role of the Lin28/let-7 axis in the development of the CSC phenotype was presented by Yu et al. [71] in a study on breast cancers. They postulated that low levels of cellular let-7 result in upregulation of HMGA2, a suppressor of differentiation, and H-RAS, an enhancer of self-renewal ability [71]. They showed that an artificial upregulation of let-7 through lentiviral infection reduced the regenerative abilities of undifferentiated cells and, subsequently, prevented tumour growth and metastasis [71]. Similar findings were reported for pancreatic cancer, where low levels of let-7 were found to be responsible for the upregulation of pluripotency genes, including Sox2, Sox9, Oct4, Nanog, and c-Myc [72]. Further evidence supporting the role of Lin28/let-7 in the development of treatment resistance through the CSC phenotype was demonstrated in breast and gastric cancer [71,73]. Overall, the Lin28/let-7 axis plays an important role in cancer stemness, and an understanding of mechanisms underlying Lin28-mediated CSC initiation might illuminate some important targets for metastatic and therapy-resistant cancer treatment.

Several known let-7-regulated signalling pathways are implicated in CSC development, including the Wnt/β-catenin, NOTCH/hedgehog, MAPK/ERK, PI3K/AKT, and STAT3/NFκB/cytokines pathways [74]. A significant amount of research on breast cancer progression linked Lin28/let-7 with the development of CSC phenotypes through the aforementioned signalling pathways. For example, breast cancer has been shown to respond to tamoxifen treatment and radiotherapy more effectively as a result of let-7c/d-induced suppression of the Wnt pathway, which stimulates CSC renewal and proliferative abilities [75,76]. Similarly, the CSC phenotype in breast cancer cells was suppressed after the attenuation of Lin28A expression caused let-7b-dependent Wnt pathway inhibition [77]. Besides breast cancer, the Wnt signalling suppressed by let-7 was shown to inhibit EMT and self-renewal phenotypes in CSCs of lung and esophageal cancers, as well as in hepatocellular carcinoma [77,78,79]. The importance of the Lin28/let-7 axis on Wnt signalling was also highlighted in a study of Wnt-activated esophageal squamous cell carcinoma (ESCC) clinical tissues. A clear correlation between metastasis, migration, and invasion with the let-7 members was observed [57].

The NOTCH/hedgehog pathway has also been indirectly linked to the stemness phenotype and metastatic abilities of cancer cells. For instance, sonic hedgehog protein, along with CSC cell surface markers, was found to be upregulated in highly metastatic pancreatic cancer [80]. Such upregulation was accompanied by low levels of let-7 members, which were shown to directly affect sonic hedgehog expression [81]. In non-small lung cancer cells (NSCLC), upregulation of let-7c, along with miR-200b, coupled with inhibition of the Hh signalling pathway resulted in a reduction of CSC markers and EMT phenotype, as well as re-sensitization to chemotherapy [82].

Aberrant regulation of the STAT3/NFkB pathway, which promotes mesenchymal and CSC phenotypes, is also related to the Lin28/let-7 axis. For example, in NSCLC, NFkB binds to Lin28B chromatin, leading to overexpression of Lin28B and subsequent suppression of let-7 [83]. Similarly, STAT3 directly stimulates Lin28A/B expression by binding to its promoter in breast cancer cells [84]. In both studies, NFkB- and STAT3-mediated activations of Lin28 enhanced EMT and CSC phenotypes. STAT3/NFkB pathway is stimulated by pro-inflammatory cytokines produced by M1 macrophages, thus mediating an immune-dependent activation of the Lin28/let-7 axis [85]. Other studies also reported that IL6 cytokines create a positive feedback loop in Lin28/let-7 axis activation through stimulation of NFkB, leading to the transformation of breast epithelial cells into CSCs. Overall, the STAT3/NFkB pathway has been found to deliver immune signals down to the Lin28/let-7 axis.

## 5. The Role of Lin28 in EMT and Metastasis

EMT, the initiation step of invasion and metastasis, is defined by the destabilization of the adherent and tight junctions that maintain the integrity and apical-basal polarity of epithelia [86,87]. The transformation of the epithelial cells into mesenchymal cells is a characteristic change when tumour cells gain an invasive phenotype and obtain motility to progress into more malignant tumours [86,87]. The epithelial cell adherent junction consists of cadherin adhesion receptors, such as N cadherin and E cadherin, which are responsible for calcium-dependent adhesion and maintenance of epithelial cell plasticity, respectively [87]. During EMT, cells lose E cadherin but gain N cadherin expression. EMT is also accompanied by several other phenotypic changes in epithelial cells, including cytoskeleton dynamics, which allow cancer cells to gain high mobility [87].

### 5.1. Lin28 Promotes EMT and Metastasis in let-7-Dependent Manner

Lin28 is one of the well-studied EMT-inducing factors [19]. Clinical evidence has shown that Lin28A and B are correlated with cancer cell invasion and migration. Early research first marked low levels of let-7 family members as a prerequisite for metastasis and one of the key drivers of mesenchymal phenotype switch in epithelial cells [88,89,90,91]. The discussed mechanisms involved loss of let-7-regulated suppression and subsequent upregulation of genes, such as E-cadherin transcriptional repressors ZEB1 and SIP1 in a breast cancer model, GAB2 and FN1 in mammary carcinoma, and developmental genes Twist and Snail in an oral squamous cell carcinoma (OSCC) model. Importantly, the key role of let-7 in EMT was further proven by reversing the EMT process through the restoration of let-7 levels [90,92]. These findings suggest that Lin28A and B are both direct negative regulators of let-7 in initiating EMT.

The direct role of Lin28A/B in the development of EMT through let-7 regulation has also been experimentally demonstrated by a series of studies. Lin28A/B was shown to be significantly upregulated in mesenchymal type cells (MDA-MB-231 and SK-3rd) compared to epithelial type cells (MCF-7 and BT-474) [20]. Furthermore, Lin28A/B was found to be responsible for higher colony formation rates and migratory abilities in breast cancer cells. In triple-negative breast cancer (TNCB) cells, overexpression of Lin28A/B lowers the level of E cadherin while increasing N cadherin, thereby evidently contributing to EMT development and conferring cancer stem cell characteristics [19]. Similarly, overexpression of Lin28B in TNCB was found to promote lung and breast cancer metastasis through the development of an immune-suppressive pre-metastatic niche marked by neutrophil recruitment and subsequent conversion of N2 phenotype [66]. In particular, Lin28B was shown to increase IL-6 and IL-10 cytokine production to polarize neutrophils into metastatic-promoting N2 phenotype-inhibiting T cells, which normally prevent malignant cell migration [66]. These studies collectively demonstrate that Lin28 initiates EMT and metastasis in breast cancer models not only by causing cell–cell adhesion loss but also by acting as an immune-suppressive agent.

Another study emphasized the role of Lin28 in EMT and metastasis by discovering Lin28 as a target of a known metastasis suppressor in PC, called Raf kinase inhibitory protein (RKIP) [67]. The study showed that RKIP indirectly suppresses Lin28A/B through mitogen-activated protein kinases (MAPK) and Myc [67]. The reduction of Lin28A/B consequently restores let-7 levels, which in turn block the expression of a metastasis-activating gene HMGA2 [67]. The dysregulation of RKIP in breast cancer cells was linked to breast cancer development and invasion through the reversion of the aforementioned cascade involving HMGA2-let-7-Lin28A/B-Myc-MAPK-RKIP [67]. Importantly, while most of the studies implicated the role of Lin28 in cancers by showing the downstream effects of Lin28, this study proved its significance by showing that Lin28 is a direct upstream regulator of RKIP.

### 5.2. Lin28 Promotes EMT and Metastasis in let-7-Independent Manner

Lin28 can also regulate cell invasion and migration independently of let-7. For example, Lin28A/B was found to act as a post-translational driver of metastatic phenotype via direct enhancement of the translation of MYCN-induced transcripts in MYCN-amplified neuroblastoma [69]. In this study, ZKD of Lin28B bound the RPS29 protein, which was shown to correlate with low patent survival, cause cell cycle arrest, and induce cancer cell morphological differentiation. Importantly, the authors showed that the loss of Lin28B reduced the metastatic potential of the tumour cells, leading to a complete reversal of stage 4 to stage 1 of neuroblastoma. This let-7-independent function of Lin28 demonstrates that Lin28 has a broad range of effectors that are highly clinically relevant.

## 6. The Role of Lin28 in Glucose Metabolism

Lin28 plays a critical role in the Warburg effect, which is defined as the reprogramming of glucose metabolism to enhance tumourigenesis [93]. Cancer cells predominantly rely on aerobic glycolysis and utilize glucose as the energy source instead of more efficient default pathways, such as the citric acid cycle and oxidative phosphorylation in mitochondria [93]. The shifts in glucose metabolism provide additional plasticity for cancer cells and even enhance therapy resistance [94]. Cancer cells require higher energy demands, not only because of the increased replication and proliferation rates but also hostile tumour microenvironments, such as changes in pH. Thus the cancer cells’ metabolic plasticity becomes critical for survival. This is even more true for cells that undergo EMT and metastasis, during which processes the cells experience higher oxidative stress and have to alter their metabolism to withstand the hurdles of metastatic spread [95].

Early evidence showed that the Lin28/let-7 axis regulates metabolism through pyruvate dehydrogenase kinase 1 (PDK1) in an oxygen- or hypoxia-inducible factor-1 (HIF-1)-independent manner [96]. Lin28B was found to control the aerobic glycolysis metabolic state and lower the microenvironment pH through the Lin28B/Myc/miR-34a-5p pathway in breast cancer stem cells (BCSCs) [97]. Lin28B-positive MDA-MB-231 and H1299 cells used a significant amount of glucose for aerobic glycolysis and produced a high concentration of lactate, while Lin28B-negative MDA-MB-231 cells had significantly lower rates of glucose uptake and lactate secretion [97]. The high level of lactate in these BSCSs promoted a low pH microenvironment that enhanced stemness properties. Furthermore, low pH increased the abundance of cells expressing high levels of aldehyde dehydrogenase 1 (ALDH1), a CSC biomarker [97]. Myc and Lin28B proteins were also significantly overexpressed in spheroid cells and ALDH-overexpressed cells under a low pH environment [97]. Moreover, the biogenesis of miR-34a-4p, a Myc-suppressed miRNA, is downregulated by Lin28B [97]. Lin28B activates Myc expression, which, in turn, blocks miR-34a-5p to induce glycolysis and inhibits oxidative phosphorylation in BCSCs. Lin28B inhibition by a small molecule C902 reduced Myc expression and lactate secretion in BCSCs. This thereby reversed pH to close to neutral and restored miR-34a-5p levels, which led to reductions in tumour spheroid formation, BCSC migration capability, and tumour sizes. Moreover, the blockade of the Lin28/Myc/miR-34a-5p axis successfully hampered pulmonary metastasis of BCSC in in vitro models [97]. Based on these findings, this study concluded that BCSC relies on aerobic glycolysis activated through the Lin28B/Myc/miR-34a-5p pathway.

Similarly, another study determined the role of the Lin28B/let-7 pathway in regulating glycolytic metabolism, ATP production, and neurosphere formation in neuroblastoma (NB) [98]. The researchers proposed a pathway whereby ornithine decarboxylase (ODC) regulates polyamine production to induce hypusination of eIF-5A, which in turn regulates the Lin28B/let-7 pathway [98]. The study showed that ODC inhibition by difluoromethylornithine (DFMO) restores ATP levels, let-7 expression, and glycolytic metabolism in various degrees depending on the NB cell lines (BE(2)-C, SMS-KCNR, and CHLA90) [98]. Lin28B and MYCN protein expression and ATP levels significantly decreased in BE(2)-C and SMS-KCNR cell lines in response to DFMO treatment [98]. Furthermore, as let-7 expression increased upon DFMO treatment, there was a decrease in glycolytic metabolism in vivo [98]. Therefore, these researchers concluded that Lin28 significantly contributes to glucose metabolism reprogramming in NB cell lines.

Although one of the main known roles of Lin28 is to promote tumourigenesis and metabolic diseases, the reprogramming of glucose metabolism by Lin28A/B brings advantages to the resistance to diabetes, skeletal muscle maintenance, and acceleration of tissue repair [99,100,101]. For instance, overexpression of Lin28A and Lin28B leads to an insulin-sensitized state, which subsequently improves glucose tolerance in Lin28A/B genetically modified aging mice. It allowed the mice to gain resistance to diet-induced diabetes under high-fat diet conditions [99]. Another study showed that Lin28A knockout transgenic mice have impaired primordial germ cell development during embryogenesis and significantly lowered fertility in the adult state. [100]. Consistently, a deficiency in Lin28A lowers the ATP/AMP ratio and NADH/NAD ratio in embryonic growth and hinders the rate of tissue repair [101]. Overall, Lin28 acts as a double sword that reprograms glucose metabolism to induce the Warburg effect in cancers and acts as a factor necessary for healthy developmental processes.

## 7. Lin28 Targets Various Messenger RNA

Besides microRNAs, Lin28 also targets messenger RNAs in a cell type-dependent and region-specific manner [102,103]. Similar to conventional mRNA binding proteins, Lin28A/B binds an array of similar RNA sequence motifs [103]. Those motifs are characterized by uridine-rich fragments surrounded by guanosine residues. Lin28A/B increases the protein levels of cell cycle regulators predominantly by stabilizing their respective mRNAs through interaction with the exonic regions and 3’ UTR of post-splicing mRNAs [103]. The conserved GGAGA sequence shared between most of the let-7 members was also found in many Lin28-targeted mRNAs [104]. By interacting with the RNA motifs in mRNAs, Lin28A/B serves as both positive and negative regulator of mRNA processing and translation. This was reported to occur through interactions with protein factors such as hnRNP F, TIA-1, FUS/TLS, and TDP-43 [104]. Interestingly, Lin28A/B autoregulates its own mRNA for translation [104]. These findings indicated that Lin28 has a broad range of functions other than controlling let-7 miRNA expression.

Moreover, Lin28 is a global regulator of different cellular compartments, including stress granules (SGs) and cytoplasmic processing bodies (P-bodies) [105]. The formation of SGs is a characteristic response of cancer cells when encountering therapy-induced stress. SGs contribute to cancer cell survival by reducing stress-related damage from oxidative, pH, and temperature fluctuations [105,106,107]. In SGs, Lin28A/B were found to bind T cell intracellular antigen 1 (TIA 1), an RNA-binding protein that stabilizes the mRNA and regulates splicing patterns [105,107]. Overexpression of TIA-1 induces SG assembly and alters gene expression and signal transduction patterns to promote cancer proliferation and metastasis [106,108]. These studies indicate that Lin28 may play a key role in regulating the survival response of tumour cells when encountering therapy-induced stress.

## 8. Lin28 and Prostate Cancer (PC)

Prostate cancer (PC) is one of the most common malignancies in men worldwide [109]. Even though PC generally has a relatively slow growth rate and low mortality rate, therapy-resistant CRPC poses a significant threat and takes thousands of lives every year [110]. These tumours fail to respond to the first-line androgen deprivation therapies due to the loss of conventional dependency on the AR signalling for growth [111]. Many tumours progress into even more aggressive states by changing cancer cell lineage and transform into therapy-induced neuroendocrine prostate cancers (t-NEPCs) that have a median survival of less than one year [112]. Thus, there is a clear need for a comprehensive understanding of the biological mechanisms underlying PC development and its progression to develop effective treatments. Lin28 has been identified as one of the most promising candidates that regulate therapy-resistant tumour progression [21]. Herein, we discuss the role of Lin28 in the development and progression of PC.

Early studies linked the role of Lin28 to the development of CRPC through the observation that AR expression was regulated by the members of let-7 through c-Myc. High levels of Lin28A-suppressed let-7 in CRPC cells were found to indirectly stimulate AR expression [113]. A follow-up study confirmed that elevated levels of let-7c in PC xenografts significantly reduced tumour growth and invasion [114]. Furthermore, PC patient tissues were found to have a higher level of Lin28A protein than the benign tissues. Borrego-Diaz et al. [51] also reported that Lin28 is an important driver of PC, whereby Lin28B regulates c-Myc expression indirectly by binding to the intermediate c-Myc miRNA suppressor miR-212. Further mechanistic studies have shown that the LNCaP PC cell lines overexpressing Lin28A rendered these cells more resistant to antiandrogens, including enzalutamide, abiraterone, and bicalutamide [115]. The resistance was attributed to the Lin28-dependent stimulation of overexpression of the AR splice variant AR-V7. Lovnicki et al. [16] found that Lin28B, along with another core pluripotency stem cell gene, Sox2, are key drivers of t-NEPC [16]. Importantly, Lin28B induced a stem-like gene network and neuroendocrine biomarkers, including HMGA2, that stimulated the transition of AdPC to more aggressive t-NEPC. Furthermore, Lin28B stimulated EMT properties, which promote t-NEPC formation, by reducing N cadherin and increasing E cadherin. Similarly, another group illustrated that upregulation of Sox2 via Lin28B induces androgen ablation-induced neuroendocrine differentiation via neuroendocrine (NE) lineage in prostate luminal cells [116]. Overall, the Lin28/let-7 pathway regulates several downstream gene networks in CRPC progression.

The expression of Lin28 is regulated by oncogenic pathways such as ESE3/EHF [117]. In normal cells, ESE3/EHF binds Lin28 promoters and inhibits its transcription, which results in an upregulation of let-7 miRNA [118]. In this way, ESE3/EHF maintains a balance between self-renewal and differentiation of normal prostatic epithelium. Significant downregulation of ESE3/EHF was reported in PC [119,120]. It was estimated that approximately 50% of PCs acquired ETS dysregulation through gene rearrangement [117]. Lin28 silencing, both in vitro and in vivo, resulted in similar phenotypes in PC cells expressing high levels of ESE3/EHF. In particular, there was a decrease in PC cell spheroid formation and self-renewal abilities, indicating that Lin28 promotes tumourigenic and CSC-like phenotypes [52]. In addition, the effects of Lin28 silencing were tested on NOD scid gamma (NSG) mice, where tumourigenic ESE3KD-PrEC cells transfected with siRNA against Lin28 caused a reduction in xenograft growth in mice [52]. These findings suggest that ESE3/EHF is a key regulator of Lin28 expression in PCa.

## 9. Therapeutic Inhibition of Lin28

Despite a widespread view that RNA-binding proteins are undruggable, there is a growing body of research on targeting Lin28A and Lin28B with small molecules [121,122,123]. These small molecules bind to the surface of either CSD or ZKD of Lin28 and prevent Lin28/let-7 interaction from increasing let-7 biogenesis in cancer cells, resulting in suppression of cancer CSC and EMT phenotypes (Table 2). Notably, the studies outlined in this section employed novel high-throughput screening techniques, including fluorescence polarization (FP) and fluorescence resonance energy transfer (FRET), which are gaining popularity in screenings for RNA-binding protein inhibitors [124,125]. These novel technologies have enabled screenings of libraries containing up to 100,000 compounds (Table 2). The basic principle of the two most commonly used techniques, FP and FRET, involves the use of fluorescently labeled macromolecules that excite light differently upon RNA–protein interactions [126]. In this way, an event of inhibition between Lin28 and let-7 RNA can be tracked.

### 9.1. Lin28 Inhibitor Screening: Fluorescence Resonance Energy Transfer

The earliest implementation of the FRET assay for Lin28 inhibitor screening yielded compound 1632, which successfully restored let-7 levels in Huh7 cells and reduced “stemness” in mouse embryonic stem cells (mESC) [136]. Compound 1632 suppressed mRNA levels of known markers of pluripotency, including Pou5f1/Oct4, Rex1, and Stella, and increased differentiation. Furthermore, the authors demonstrated that 1632 effectively inhibited clonogenic growth in 22Rv1, PC3, DU145, and Huh7 cell lines and induced dose-dependent reduction of tumour-sphere formation in 22Rv1 and Huh7 cells. Subsequent studies demonstrated that **1632** downregulates Lin28 mRNA levels, increases let-7 miRNA levels, and, importantly, suppresses cell surface expression of PD-L1, one of the key proteins involved in immune evasion of cancer cells [135]. 1632 was reported to block glycolytic, tumour sphere-forming, and migratory capacities in MDA-MB-231 and H1299 cell lines, as well as reduce the stemness phenotype [97]. The pharmacological effects of 1632 are also observed in vivo, as it inhibited tumour growth and metastasis in immunodeficient orthotopic mouse models of MDA-MB-231 human breast cancer cells. Additionally, compound 1632 suppresses cell growth and clonogenicity in acute myeloid leukemia cell lines and tissue samples [62]. The mechanisms involved in the inhibition of the NFkB pathway induced leukemic stem cells and cell cycle arrest at the G1/S phase. Collectively, these studies demonstrated that compound 1632 is a promising drug candidate for Lin28 overexpressing tumours.

Other compounds identified by FRET also showed some promising activity in vitro. In particular, compound KCB3602 was found to increase let-7 levels in human choriocarcinoma JAR cells and, subsequently, downregulate gene products of the key drivers of CSC, including HMGA, IGF2, IMR1, Ras, and Lin28A [128]. Furthermore, KCB3602 effectively suppressed the sphere-like growth of cancer cells, a hallmark of CSCs. The authors confirmed the on-target effects using surface plasmon resonance (SPR) assays and showed that KCB3602 does not impair cell growth in a non-cancerous cell line. Another small molecule, benzopyranylpyrazole-based compound (1), was found to upregulate let-7 levels in JAR and PA-1 cell lines [129]. It also downregulated the expression of let-7 target genes, including HMGA2, c-Myc, and Ras. These compounds proved that the pharmacological inhibition of Lin28 suppresses CSC gene signatures. However, further investigations are necessary to determine if these inhibitory effects can be advanced to in vivo studies. 

### 9.2. Lin28 Inhibitor Screening: Fluorescence Polarization

Another commonly used technique is the FP assay, which was first used to screen for Lin28 inhibitors by Lightfoot and colleagues in 2016 [133]. The authors reported two small molecule inhibitors, SB/ZW/0065 and 6-hydroxy-DL-DOPA, had low micromolar activity in FP assays. Electrophoresis mobility shift assay (EMSA) was then used to confirm the disruption of Lin28/pre-let-7g complex. Furthermore, these molecules were shown to partially restore Dicer cleavage activity of pre-let-7g in vitro but failed to restore let-7g levels in P19 embryonal carcinoma cells. Later, Wang et al. [133] also employed FP screening of a large library and presented two small molecule inhibitors—namely, LI71 and TPEN—that targeted the CSD and ZKD domains of Lin28A and Lin28B, respectively. Importantly, the authors confirmed the direct binding of these molecules to their respective target domains—i.e., CSD and ZKD—with protein nuclear magnetic resonance (NMR) spectroscopy. LI71 treatment upregulated various members of the let-7 family in leukemia cell line K562 and mouse embryonic stem cells, while TPEN was generally toxic in vitro. Using LI71 as a scaffold, further modifications resulted in derivatives with stronger potency in FP and EMSA assays [131]. They performed structure–activity relationship (SAR) and docking-based binding mode analyses and proposed that the binding mode features are necessary to inhibit protein–RNA interaction at the Lin28 CSD interface. These features included a π–cation interaction with Lys102 and several hydrogen bonds with Lys78, Lys98, and Ala101. Interestingly, another study performed a similar series of experiments (i.e., FP HTS, SAR, docking) and presented a new class of trisubstituted pyrrolinone inhibitors, including the hit compound C902. This work highlighted the importance of Lys102 for the successful inhibition of Lin28 CSD [130].

### 9.3. Lin28 Inhibitor Screening: Other Methods 

A few other studies have focused on the development of novel experimental high-throughput techniques in the context of Lin28/let-7 interaction inhibition. For instance, catalytic enzyme-linked click chemistry assay (cat-ELCCA) enabled the identification of two *N*,*N*′-(1,2-phenylene)-dibenzenesulfonamide derivatives, CCG-233094 and CCG-234459 [134]. The compounds were shown to inhibit Dicer-mediated cleavage of pre-let-7d and on-target effects confirmed by SPR. Another study focused on the development of a fluorescence intensity-based (FL) assay to screen for potential Lin28/pre-let-7 interaction disruptors [137]. The authors validated their method by identifying a novel scaffold, KCB170522, which was also active in EMSA and showed an expected increase in cellular let-7 levels. Besides small molecules, an epitope inhibitor named Nb-S2A4 was developed and shown to indirectly impact ZKD through recognition of N-terminal TUT4 and disruption of its association with Lin28B ZKD/pre-let-7 complex [132]. Nb-S2A4 prevents pre-let-7 degradation by allowing oligouridylation and monouridylation activities of TUT4 without stopping the formation of the Lin28B/pre-let-7 complex. In summary, several bioassays have been established to screen and validate Lin28 inhibitors that can block Lin28/let-7 interactions and suppress the CSC and EMT phenotypes of cancer cells. Although several Lin28 compounds had been discovered and developed, none have reached clinical trials, and more efforts in the field of research are needed.

## 10. Conclusions

Both Lin28A and Lin28B play important roles in cancer development and progression. As CSC promoters, they induce the stemness phenotype and drive the self-renewal and clonogenic capabilities of cancer cells. The Lin28-mediated stemness gene network confers therapy resistance in various cell and xenograft models. It promotes cancer cell migration and invasion, leading to cancer progression to more aggressive stages. Importantly, Lin28 overexpression is observed in a wide range of cancer types, highlighting that it could be used as a common prognostic marker and therapeutic target of cancer.

The development of Lin28 inhibitors as therapies is still at its early stage, as most studies focus on the screening techniques and aim to provide proof-of-concept for Lin28 inhibition. However, a few promising molecules have emerged and been shown to successfully restore let-7 levels and downregulate downstream oncogenes. Furthermore, some studies showed that the pharmacological effects of Lin28 modulation result in the suppression of tumour growth in animal models. Thus, the development of Lin28-targeted drugs presents an attractive strategy for cancer treatment, especially in highly aggressive metastatic forms. The rise of computational drug discovery should aid the development of such drugs. The largest performed screening for Lin28 inhibitors covered around 100,000 compounds, whereas the available chemical space counts in the billions nowadays. Thus, virtual screening coupled with novel in vitro screening techniques should make it possible to explore the vast chemical space and find safe and efficacious small molecule inhibitors of Lin28 protein.

## Figures and Tables

**Figure 1 cancers-14-04640-f001:**
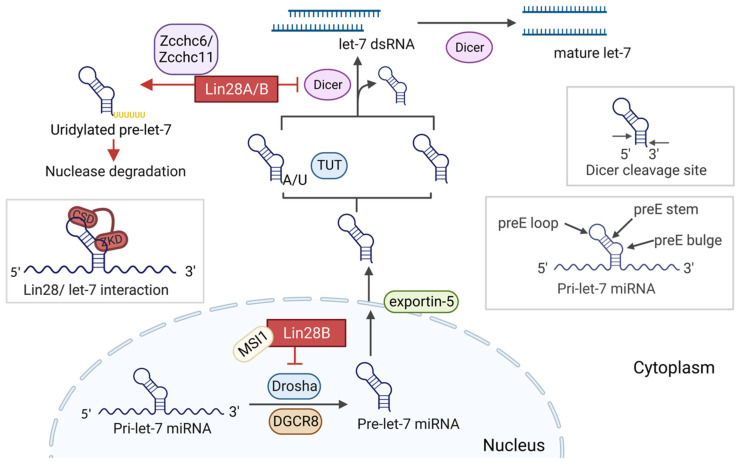
The biogenesis of let-7 microRNA. In the absence of Lin28B, pri-let-7s are cleaved into pre-let-7 miRNAs and then converted to mature let-7 with the aid of Dicer in both cytoplasm and nucleus. The stages at which Lin28A and/or B inhibit let-7 maturation are marked. The lower right square shows the structure of pri-let-7. The upper right square demonstrates the site where Dicer cleaves pre-let-7. The left square shows the bipartite binding of Lin28A/B to pri/pre-let-7 members. Created with BioRender.com.

**Table 1 cancers-14-04640-t001:** Lin28 roles in cancers. Lin28A and B expression in human cancer cell lines were confirmed from the Cancer Cell Line Encyclopedia (CCLC) Dependency Map database.

Primary Body Site	Tumour Name	Samples Studied	Lin28A or B Expression	Results	Ref.
Lung	Non-Small cell lung cancer (NSCLC)	Lung tissues Cell lines: A549, NHBE	A549: both NHBE: n/a	(1) Lin28B and metabolic enzyme glycine decarboxylase (GLDC) ↑ in tumour-initiating cancer cells (TICs) are necessary and sufficient to induce tumour sphere formation	[47]
	Small cell lung cancer (SCLC)	Cell lines: NCI-H446	NCI-H446: both	(1) Lin28A ↑ with c-Myc ↑ causes pri-let-7 ↓and mature let-7 ↓ (2) Lin28A inhibition triggers cell cycle arrest and ↓ cell growth (3) Lin28B LoF results in CDC25A ↓, an activator of G1 to S phase transformation, thus preventing cell division	[48]
Brain	Glioblastoma multiforme (GBM)	Glioma tissues Cell lines: U251, U373	All cell lines: Lin28A	(1) Lin28A ↑ leads to poor diagnosis and ↓ survival rate (2) Lin28A ↓ induces cell cycle arrest in G1 phase, ↑ apoptosis, and ↓ colony count	[49]
	Atypical teratoid rhabdoid tumour (AT/RT)	AT/RT primary tumour tissues Cell lines: BT12/37, CHLA-06/04/06	All cell lines: both	(1) High levels of Lin28A and Lin28B mRNA coupled with low levels of let-7 in the majority of tumours (2) Lin28A KO ↑ let-7 mRNA, downregulates KRAS mRNA, reduces cell growth and proliferation, ↓ clonogenicity, and induces apoptosis in cells (3) Lin28A ↓ in xenograft results in a two-fold ↑ in median survival rate	[50]
Prostate	Androgen-independent prostate cancer (AIPC)	Cell lines: VCaP, LNCaP, PC3, Du145	VCap: both LNCap: Lin28B PC3: Lin28B Du145: Lin28B	(1) Lin28B positively correlates with c-Myc protein but not mRNA level (2) Lin28B negatively correlates with miR-212, a tumour suppressor (3) Computational predictions suggest that c-Myc protein activates Lin28B, which in turn ↓ miR212 miRNA activity	[51]
	Therapy-induced neuroendocrine prostate cancer (t-NEPC)	Cell lines: LNCaP, C4-2, RWPE-1, 22RV1, PC-3, VCaP, Du145, NCI-H660/H82/H69, LASCPC	LNCap: Lin28B C4-2: n/a RWPE-1: n/a 22RV1: n/a PC-3: n/a VCaP: both Du145: Lin28B NCL-H660/H82/H69: Lin28B LASCPC: Lin28B	(1) Lin28B ↑ and Sox2 ↑ mRNAs and protein levels in tumours, xenografts, and cells (2) Let-7d ↓ results in HMGA2 ↑ and Sox2 ↑ (3) Lin28B induces cancer stem cell and neuroendocrine biomarkers	[16]
	Prostate cancer (PC)	Cell lines: PrEC, RPWE-1, LNCaP, Du145, PC3	All cell lines: both	(1) ESE3/EHF binds Lin28A and B promoter, deactivating its expression and, thereby, ↑ let-7 miRNA levels (2) Lin28A and B inhibition resembles the effects of ESE3/EHF, as it ↓ CSC phenotype	[52]
Gastrointestinal	Colon carcinoma	Colon cancer tissues Cell lines: SW480, HCT116	All cell lines: both	(1) Lin28B ↑ in colon cancer tissues ↑ metastasis and correlates with poor patient survival rate (2) Lin28B inhibition leads to ↓ migration and ↑ drug-induced cytotoxicity in cells	[53]
	Colon adenocarcinoma	Colon carcinoma tissues Cell lines: DLD-1, LoVo	DLD-1: Lin28B LoVo: both	(1) Lin28B ↑ reduces patient survival, ↑ tumour resistance and chances of relapse (2) Lin28B ↑ LGR4 and PROM1 colonic cancer cell markers in vivo	[54]
	Gastric adenocarcinoma (GAC)	Gastric cancer tissues	Lin28B	(1) Lin28B ↑ in GAC tissues (2) Lin28B expression level is positively correlated with clinicopathological parameters of GAC patients (i.e., lymph node status, TNM, tumour invasion, and GAC cell differentiation) and negatively correlated with survival rates	[55]
	Colorectal cancer (CRC)	Colon adenocarcinoma tissues	Both	(1) Lin28A/B levels are not related to liver metastasis development (2) Lin28A/B expression is negatively correlated with Sox2 expression	[56]
	Wnt-activated esophageal squamous cell carcinoma (ESCC)	ESCC tissues Cell lines: TE-1, ECA109, KYSE-150	TE-1: both ECA109: Lin28A KYSE-150: both	(1) Let-7a ↓correlates with metastasis and cancer recurrence (2) Let-7a-mimic and Lin28A/B silencing reduce EMT Let-7a ↓ reduces protein levels of EMT divers Snail and SLUG (3) Wnt/β-catenin ↑ increases the expression of its Lin28A/B	[57]
	Oral squamous cell carcinoma (OSCC)	OSCC tissues Cell lines: SCC9, SCC15, SCC25	SCC9: both SCC15: both SCC25: Lin28B	(1) Lin28B expression correlates with earlier disease recurrence and promotes cancer cell migration (2) Lin28B, but not Lin28A, is associated with poor patient prognosis (3) Lin28B ↑ migration, invasion, proliferation, and clonogenicity (4) Lin28B ↑ IL-6, HMGA2, Snail, Twist, VEGF, and Survivin expression	[58]
Kidney	Renal cell carcinoma (RCC)	Nephrectomy tissues	Lin28A	(1) Lin28A expression is normal	[59]
	Wilms’ tumour	Kidney tissues of mice embryo	Embryonic stem cells: both	(1) Lin28A and B ↑ blocks differentiation of embryonic kidney cells, transforming them into nephrogenic progenitors that initiate Wilms tumour (2) Lin28B/let-7 pathway promotes glomerulus-like structure tumour formation	[60]
Circulation	Mixed-lineage leukemia (MLL)	Bone marrow tissue MLL-AF6/9/10/ENL/Gas7	Lin28B	(1) Lin28B ↑ due to c-Myc ↑ (2) Lin28B inhibits let-7a/g and ↑ tumour growth (3) Let-7 levels’ restoration results in differentiation of leukemia blasts	[61]
	Acute myeloid leukemia (AML)	AML xenograft mice model AML cell lines	Lin28A and B	(1) A small molecule inhibitor (1638) of Lin28 ↓ cell growth and clonogenicity (2) In xenograft models, 1638 leads to ↑ let-7 levels, ↓ LCS counts and tumour growth	[62]
	Juvenile myelomonocytic leukemia (JMML) subtype	JMML tissues	Lin28B	(1) Lin28B is a characteristic of a novel molecular subgroup of JMML (2) Lin28B correlates with HbF, a prognostic JMML marker (3) Lin28B is a marker of poor survival	[63]
Liver	Hepatocellular carcinoma (HCC)	HCC tissues Cell lines: Hek293, Cos-7 HepG2/3B, Li-7, HLE, Huh6/7, MCF-7	All cell lines: Lin28B Huh6/7: both	(1) First report of Lin28B (2) Lin28B ↑ in HCC cell lines and tissue samples (3) Lin28B localizes in the cytoplasm (4) Lin28B lacking CSD domain is expressed in normal liver cells	[45]
		Cell lines: HeLa, HepG2, Huh7, Hep3B	HeLa: Lin28B HepG2: both Huh7: both Hep3B: both	(1) Lin28B ↓ let-7 biogenesis by inhibiting its maturation through 3’ uridylation that blocks Dicer processing	[32]
	Hepatoblastoma (HB)	Mouse models	Lin28B	(1) Lin28B alone is sufficient to induce liver tumourigenesis in mice model (2) Lin28B ↑ in Myc-driven mouse models (3) Lin28B directly ↑ an lgf2 mRNA binding protein that ↑ tumour growth	[46]
Ovarian	Ovarian carcinoma	A panel of 527 human	Both	(1) Lin28A and Lin28B ↑ in ovarian carcinoma histological grade 2 or 3	[25]
	Epithelial ovarian cancer	Epithelial ovarian cancer patient tissues	Both	(1) Lin28B ↑ in patients leads to short progression-free and low survival rates (2) Lin28B blocks let-7a maturation and positively correlates with pri/pre-let-7a (3) A positive correlation between Lin28B and insulin-like growth factor II (IGF-II) may contribute to the adverse effect in ovarian cancer	[64]
	Ovarian primitive germ cell tumours (GCTs)	Ovarian cancer tumour tissues	Both	(1) Lin28A ↑ significantly contributes to primitive ovarian GCTs (primary and metastatic dysgerminomas, gonadoblastoma, Yolk Sac tumour, and embryonal carcinoma), immature teratomas, and neuroepithelial tissues (2) Lin28A can be used as a diagnostic marker for GCTs	[26]
Testicle	Testicular germ cell tumours (GCTs)	Testicular cancer tumour tissues	Both	(1) Lin28A and Lin28B are strongly detected and have specificity in metastatic testicular GCTs, including classic seminoma, embryonal carcinoma, and yolk sac tumour (2) Lin28A, Lin28B, and SALL4 act as sensitive markers for extragonadal yolk sac tumours	[27]
Breast	Breast cancers	Breast cancer tissues HB22, T47D, MCF7, Bcap-37, SK-BR-3, MDA-231	HB22: n/a T47D: both MCF7: Lin28A Bcap-37: SK-BR-3: Lin28A MDA-231: n/a	(1) Lin28A/B ↑ confers resistance to paclitaxel (2) Lin28A/B KO causes ↑ sensitivity to paclitaxel in T47D cell line (3) Lin28A/B ↑ in breast cancer tissues of relapsed cancer (4) Lin28A/B induces p21 and Rb expression	[65]
	Triple-negative breast cancer subtype	Primary breast cancer tissue 4TO7, MDA-MB-231, 293 T	All cell lines: Lin28B	(1) Lin28B ↑ pre-metastatic genes in lung tissues, facilitating neutrophil accumulation, N2 conversion, and the development of the immune-suppressive pre-metastatic niche (2) Let-7s exosomes ↓ in pre-metastatic niches ↑ IL-6 and IL-10, contributing to the incidence of lung metastasis and increased tumour size (3) Lin28B and let-7s are markers for poor survival and lung metastasis	[66]
	Breast cancer	Mouse models MDA-MB-231, MCF10A, bone (1833) and lung (4175) metastatic breast cancer cells	MDA-MB-231: Lin28B MCF10A: both 1833: both 4175: both	(1) The dysfunction of Raf kinase inhibitor protein (RKIP), a metastasis suppressor, facilitates Myc binding to the Lin28A and B promoter (2) Lin28A and B suppression leads to ↑ let-7, inhibits HMGA2 (an activator of metastatic genes), and ↓ bone metastasis	[67]
Lymph Node	Papillary thyroid carcinoma (PTC)	PTC tissues	Both	(1) Lin28A and B was expressed in 40.5% of PTC samples (2) Patients with Lin28A and B expression had larger tumour size, more frequent lymph node metastasis, and aggressive tumour characteristics (3) Lin28A and B serves as a prognostic marker in PTC	[68]
Spinal Cord	MYCN-amplified neuroblastoma	CHP-212, SK-N-AS, SY5Y, BE2C, Kelly	CHP-212: both SK-N-AS: Lin28B SY5Y: Lin28B BE2C: Lin28B Kelly: Lin28B	(1) Lin28B ↑ cell migration in vitro and metastatic abilities in vivo (2) Lin28B KO ↓ tumour size in mice xenograft (3) Lin28B mediates liver metastasis MYCN amplification significantly induces Lin28B expression	[69]
Bladder	Bladder cancer (BLCa)	BLCa tissues Cell lines: T24, UM-UC-3, J82, SV-HUC-1	All cell lines: both	(1) LINC01451 IncRNA directly binds to the promoters of Lin28A and Lin28B in tumour tissues (2) LINC01451 ↑ EMT by ↓ biomarkers of TGF-β/Smad signalling pathways through Lin28A/B regulation	[70]

↑—increase; ↓—decrease

**Table 2 cancers-14-04640-t002:** Lin28 inhibitors. Lin28A and B expression in human cancer cell lines was confirmed from the Cancer Cell Line Encyclopedia (CCLC) Dependency Map database.

Target Domain	Lin28 Inhibitor	Identification Method	Activities	Ref.
Cold shock domain (CSD)	LI71 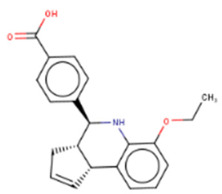	FP-based HTS of 101,017 compounds from 17 libraries	(1) Oligouridylation TUT IC50 = 27 μM (2) FP IC50 = 7 μM (3) Direct binding: NMR spectroscopy (4) Let-7 qPCR: 2–5 times (let-7b/c/d/f/g/i) in K562 (Lin28B^+^) at 100 μM 3–7 times (let-7a/b/c/d/e/f/g/i) in DKO+A (Lin28A^+^) mESCs at 100 μM	[127]
	KCB3602 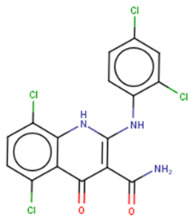	FRET-based HTS of 8400 compounds from Korea Chemical Bank (KCB)	(1) EMSA IC50 = 4.8 μM (2) Direct binding: SPR kD = 5.9 ± 2.3 μM (3) Let-7 qRT-PCR: 1.5–2.5 times (let-7a/b/d/f/g/i/miR-98) in JAR cells (Lin28A^+^, Lin28B^+^) at 10 μM	[128]
	Compound **1** 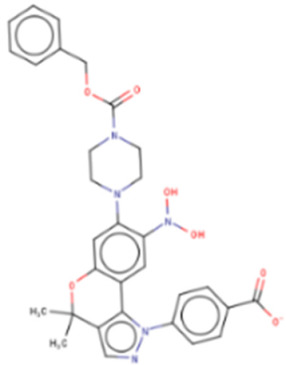	FRET-based HTS of 4500 drug-like compounds	(1) FRET IC50 = 4.03 μM (2) EMSA IC50 = 6.75–11.8 μM (Lin28A/B: let-7a-1/g) (3) Direct binding: SPR kD = 3.51 μM (4) Let-7 qPCR: ~1.5–2.5 times (let-7a/d/f/g/i/miR-98) in JAR cells (Lin28A^+^, Lin28B^+^) at 40 μM ~1.5–2.3 times (let-7a/g) in PA-1 cells (Lin28A^+^, Lin28B^+^) at 40 μM	[129]
	C902 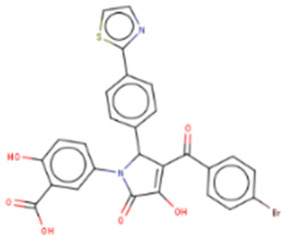	FP-based HTS of 15,000 natural product-inspired compounds	(1) FP IC50 = 5.0 μM (2) Let-7 RT-qPCR: ~2 times (let-7a/g) in JAR cells (Lin28A^+^, Lin28B^+^) at 20 μM	[130]
	GG-43 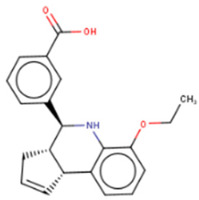	Custom chemistry modifications of LI71	(1) EMSA IC50 = 21.9 μM (LI71 EMSA IC50 = 41.6 μM) (2) FP IC50 = 4 μM (LI71 FP IC50 = 7 μM)	[131]
Zinc finger domain (ZKD)	TPEN 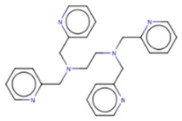	Fluorescence polarization	(1) FP IC50 = 2.5 μM(2) Toxic in Lin28^+^ mESCs and Lin28^-^ HeLa cells (Lin28B^+^) (3) Direct binding: NMR spectroscopy	[127]
	Nb-S2A4	Functional Epitope Nanobody Selection Platform	(1) Oligouridylation TUT IC50 = 0.2 μM (2) Let-7 qPCR: 3–12 times (let-7b/c/d/e/f/g/i/miR-20b) in HeLa cells (Lin28B^+^)	[132]
Unknown	SB/ZW/0065 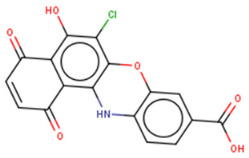	FP-based HTS of 2,768 compounds from Sigma LOPAC1280 library, NCI diversity set II, and a targeted nucleic acid structure library	(1) FP IC50 = 7.05± 0.13 μM	[133]
	6-hydroxy-DL-DOPA 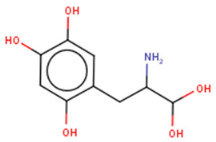		(1) FP IC50 = 4.71 ± 0.16 μM	
	CCG-234459 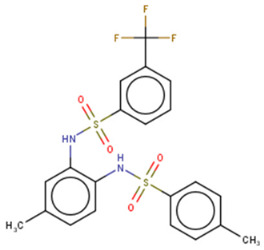	Cat-ELCCA-based HTS of 127,007 compounds from LOPAC, Prestwick, Maybridge, ChemDiv, and UMich libraries	(1) Cat-ELCCA IC50 = 8.3 μM	[134]
	CCG-233094 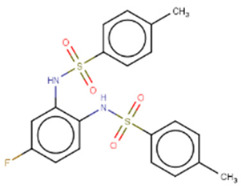		(1) Cat-ELCCA IC50 = 10.3 μM	
	**1632** 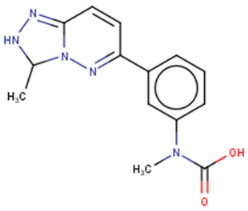	FRET-based HTS of 16,000 compounds from Maybridge Hitfinder library	(1) ELISA IC50 = 8 μM (2) On target: SPR (3) Clonogenic assay: GI50 = 20–80 μM (4) Tumour sphere-formation assay: GI50 = ~26 μM (5) Let-7 RT-qPCR: 1.2–3 times (let-7a/f/g) in Huh7 cells (Lin28A^+^, Lin28B^+^) at 60 μM 1.4–2 times (let-7a/g) in JAR cells (Lin28A^+^, Lin28B^+^) at 20 μM	[97,135,136]
	KCB170522 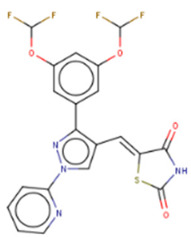	Fluorescence intensity-based (FL)-based HTS of Korea Chemical Bank (KCB), natural products, and drug-like libraries	(1) FL IC50 = 9.55(2) EMSA IC50 = 12.8 μM (3) Let-7 RT-qPCR: - 1.4–2 times (let-7a/g) in JAR cells (Lin28A^+^, Lin28B^+^) at 20 μM	[137]

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
