# Peer review of "Lin28 Regulates Cancer Cell Stemness for Tumour Progression"

_cancers, 2022, doi:10.3390/cancers14194640_

Round 1
Reviewer 1 Report
The authors reviewed all the mechanisms and aspects related to cancer cell stem-ness and tumour progression by Lin28. Overall, it is very well written review article. Except few minor mistakes as below, the article looks great.
1. Line 25 mentions “message RNAs”, it should be messenger RNAs?
2. Line 98 mentions “Lin29A”, it should be Lin28A
Author Response
Dear reviewer,
We appreciate your feedback.
We addressed your comments.
Line 25 - changed 'message' to 'messenger'
Line 98 - fixed the type and put Lin28 instead of Lin29
Reviewer 2 Report
Major points
Throughout the text, the authors need to be more clear about whether they are referring to Lin28A or Lin28B or both, as these proteins have distinct cellular localizations that impact how potentially each regulates let-7 microRNA biogenesis [see Figure 1, Page 4 Line 136, more broadly section 2.3, Table 2 (for example K562 cells are LIN28B-positive), the neuroblastoma cell lines are LIN28B, etc…]. LIN28 was the prior official gene name for LIN28A that can also be confusing if using Lin28 to refer to both genes. In cancer, it is LIN28B that is more commonly overexpressed than LIN28A. LIN28A and LIN28B tend to have mutually exclusive expression. Notable exceptions to the mutual exclusivity include but are not limited to germ cell tumors, teratomas, and ovarian cancer.
In terms of rigor of the literature review:
1. I would use the CCLE database to double check cell lines to determine whether or not they express LIN28A and/or LIN28B. I think that you will find in some referenced papers/cases that the experimental cell line expresses neither. Although there may be clonal differences between CCLE and what was used in the paper, the lack of a Western Blot is often telling in some papers.
2. I would be reluctant to consider any pan-LIN28AB antibodies as legitimate. I think some may get confused that LIN28B typically is a doublet. Of note, LIN28A and LIN28B are different sizes, so any pan-LIN28AB antibody where they claim a single band represent total LIN28AB can not be correct if they claim both genes are expressed.
On Page 5 Section 3, I would reference the genetically engineered mouse models that demonstrate that LIN28A and/or LIN28B overexpression is necessary and sufficient to cause cancer in some cases while almost always making outcomes worse in mouse models of several cancer types.
Minor points
1. Throughout the text, “none CSC” is used when it should read “non-CSC”.
2. Page 2, Line 82 I think the authors meant to refer to the four proteins as reprogramming factors for induced pluripotent stem cells not core CSC regulators.
3. Page 3 Line 98 The Lin28 genes are paralogs not isoforms.
4. Page 3 Line 106 Instead of precursors, I would say immature forms as it was reported that LIN28B may bind to pri-let-7 in addition to pre-let-7.
5. Page 3 Line 112 Are the stem region always 33bp for all let-7 family members. Please double check as this is most likely not true?
6. Page 3 Line 113 Do you need to reference Triboulet paper here as one let-7 bypasses LIN28-mediated repression.
7. Page 3 Line 120 “prevent” is too strong of a word maybe “impairs” would be better.
8. Page 3 Line 120 Drosha is nuclear. I think you meant to say the cytoplasmic DICER complex.
9. Page 3 124 I would simply say that one strand of the microRNA duplex is preferentially incorporated into the RISC complex.
10. Page 4 Line 157 What is the evidence that Lin28A is nuclear during the cell cycle?
11. Page 11 Line 304 Rewrite for clarity.
112. Page 12 Line 339 I am not sure that you can conclude this as it currently written. Are you considering single or double knockouts? Double knockouts were mid-gestation lethal. For Lin28B-nulls, growth retardation was only seen in males. Lin28A-nulls were perinatal lethal.
113. Page 11 Line 343 In adults, LIN28A and LIN28B has a very narrow expression profile (placenta, testis, ovaries, etc..). What diseases in particular is expression of either protein needed to prevent disease?
14. I would add a second reference for TUT4 & LIN28A mediating uridylation of pre-let-7 (Hagan, et al. NSMB Lin28 recruits the TUTase Zcchc11 to inhibit let-7 maturation in mouse embryonic stem cells)
15. Throughout the text, “in vivo” and “in vitro” should be italicized.
Author Response
Dear Reviewer,
We appreciate your feedback and we addressed your comments.
Major points:
1) We went through the manuscript and specified whether we refer to Lin28A, Lin28B or both. We also used CCLE database to confirm which cell lines contains which paralog of Lin28. We updates both tables and the figure with the specifications.
2) Assuming that the reviewer refers to the epitope inhibitor named Nb-S2A4, we specified that the authors designed it for Lin28B.
3) We thank the reviewer for the valuable suggestion. We added the information to section 3.
Minor points.
1,2,3,4,7,8,9,15 - all the wording or term use changes suggested by the reviewer were implemented. The words “in vivo” and “in vitro” were italicized.
5 - We removed this specification at all.
6 - The Triboulet paper is referenced later in the text.
10 - The reviewer correctly pointed that the paper stating the cell-cycle dependent localization was not referenced. We added that paper instead (Guo 2006)
11 - We rephrased the sentence to add clarity.
12 - We clarified the finding of the cited paper.
13 - We changed the sentence to reflect the importance of Lin28 during the developmental process rather than 'diseases'.
14 - We added the mentioned reference